# Effects of Plant Meristem-Cell-Based Cosmetics on Menopausal Skin: Clinical Data and Mechanisms

**DOI:** 10.3390/biom14091176

**Published:** 2024-09-19

**Authors:** Liudmila Korkina, Zaira Kharaeva, Albina Shokarova, Elena Barokova, Wolfgang Mayer, Ilya Trakhtman, Roberto Dal Toso, Chiara De Luca

**Affiliations:** 1Centre for Innovative Biotechnological Investigations Nanolab (CIBI-NANOLAB), 119991 Moscow, Russia; 2R&D Department, Swiss Dekotra GmbH, 8001 Zurich, Switzerland; trakhtman@dekotra.com; 3Immunology, Microbiology and Virology Department, Kabardino-Balkar State Medical University, 360040 Nal’chik, Russia; irafe@yandex.ru (Z.K.); albinashokarova91@gmail.com (A.S.); barokova@mail.ru (E.B.); 4R&D Department, Medena AG, 8910 Affoltern-am-Albis, Switzerland; wolfgang.mayer@medena.ch (W.M.); chiara.deluca@medena.ch (C.D.L.); 5Department of Life Sciences and Biotechnologies, University of Ferrara, 44121 Ferrara, Italy; dltrrt@unife.it

**Keywords:** asiaticoside, collagen, cosmetics, echinacoside, leontopodic acid, menopause, plant meristem cells, MDA, skin ageing, verbascoside

## Abstract

A randomised open clinical/laboratory study was performed to evaluate the safety and cosmetic efficacy of facial cosmetics for females during the menopausal period. The cosmetics contain active ingredients of meristem cells derived from the medicinal plants *Leontopodium alpinum*, *Buddeleja davidii*, *Centella asiatica*, and *Echinacea angustifolia*. Recently, the major bioactive molecules of these medicinal plants (leontopodic acid, verbascoside, asiaticoside, and echinacoside, respectively) have been thoroughly evaluated in vitro for molecular pathways and cellular mechanisms and their preventive/curative effects on human skin cells exposed to factors promoting premature skin ageing and cellular senescence. Nevertheless, clinical data on their safety/efficacy to ageing human skin are scarce. This clinical study enrolled 104 Caucasian females in pre-menopause, menopause, or post-menopause periods. They applied cosmetic serums daily for 1 month. Questionnaires and instrumental and biochemical methods were used to assess dermatological/ophthalmological safety and cosmetic efficacy through changes of the skin physiology markers characteristic of ageing/menopause (elasticity, barrier functions, moisture, sebum, ultrasonic properties, and collagen content and structure). Quantitative microbiological tests were carried out for skin microbiota fluctuations. Data showed that the cosmetics were safe, and they shifted the skin physiology parameters to a younger biological age, enhanced collagen synthesis, inhibited lipid peroxidation, and favoured normal microbiota.

## 1. Introduction

Women live approximately one-third of their lives in the pre-menopausal, menopausal, and post-menopausal states. Hormonal changes that begin during the menopausal transition affect many biological systems. Accordingly, the signs and symptoms of menopause include central nervous system-related disorders; metabolic, weight, cardiovascular, and musculoskeletal changes; urogenital epithelium and skin atrophy; and sexual dysfunction [1]. The physiological basis of these manifestations is emerging as complex and related, but not limited, to oestrogen deprivation [2,3]. Post-menopause physiological changes are particularly noticeable in the skin, which loses structural architecture and increases its sensitivity to damage [4]. Skin is one of the largest organs of the body and is significantly affected by ageing and menopause, since the oestrogen receptor is located on all cutaneous cells [3]. Therefore, dermal cellular metabolism is influenced by the deficit of oestrogen during menopause, leading to changes in collagen content and most importantly in water content. Skin collagen changes lead to diminished elasticity, dryness, and strength alteration characteristic for ageing menopausal skin [3]. Skin and mucosal menopausal symptoms include dryness and pruritus, thinning and atrophy, wrinkles and sagging, poor wound healing, and reduced vascularity [5]. The menopausal skin is extremely susceptible to microbial infections, with manifestations in the form of menopause-connected acne [5], and is also prone to chronic inflammatory skin diseases, for example, psoriasis [6]. Since collagen atrophy is a major factor in skin ageing, the aged skin is characterised by a progressive increase in extensibility and a reduction in elasticity [6].

Recent scientific discoveries have provided more insight into specific oxidative reactions that cause and accompany menopause-connected age-dependent changes in skin structure and functions. Additionally, several other processes are known which contribute to intrinsic, hormone-deficiency-induced/-maintained chronological skin ageing, such as the accumulation of advanced glycation products [7]; chronic, sterile, low-intensity inflammation described as “inflammaging” [8,9]; as well as skin microbial flora (microbiota) dysbalances [10]. A further feature of ageing skin is the accumulation in the structural skin proteins of heavy isotopes of the water molecule, such as D, deuterium (one of the heavy isotopes of H, hydrogen), and O, oxygen-18 [11]. At some point in human life, intrinsic chronological ageing is remarkably accelerated by extrinsic environment/lifestyle/diet-driven premature ageing that concurs with determining the “biological age” of the skin, as quantified by a number of objectively measurable markers, including dermal/epidermal thickness, tone, wrinkles, moisture, and elasticity.

As a therapeutic/preventive approach to alleviate menopause-related skin problems, hormone replacement therapy has been proposed and implemented [1,2,3,5,12,13,14], notwithstanding the clear evidence of compelling risks connected to long-term treatment, including carcinogenic effects [14]. Therefore, generalised hormone replacement is no longer suitable to treat merely aesthetic skin problems connected with menopause. The development and application of safe and efficient nutraceutical and cosmeceutical preparations capable of preventing/combatting skin defects induced by an age-dependent hormonal deficiency have been growing exponentially in the last decade [7,15,16].

Several years ago, our group proposed an array of active substances derived from cultivated meristem cells as promising, clinically effective, contamination-free, and ecologically friendly agents for anti-age cosmetics [15,16,17,18]. During the last two decades, these biomolecules have been thoroughly investigated, leading to the clarification of the molecular pathways and cellular mechanisms underlying their action towards human skin cells (see Table 1 for references). Here, we have carried out an open, randomised, case–controlled clinical trial on four cosmetic face serums based on actives contained in the concentrated meristem cell homogenates of *Leontopodium alpinum*, *Buddeleja davidii*, *Centella asiatica*, and *Echinacea angustifolia*. The major biologically active components of these cultured plant cell concentrates targeting human skin cells in vitro and in vivo are presented in Table 1. All these actives belong to higher plant secondary metabolites: low-molecular-weight substances with regulatory, adaptogenic, and protective properties.

Their chemical structures are shown in Figure 1.

Major goals of the study were to prove the dermatological and ophthalmological safety of the Infinity Line cosmetic serums containing optimal amounts of these substances synthesised in the corresponding meristem cell cultures, as well as to demonstrate the cosmetic efficacy, if any, of the compositions towards relevant physiological, biochemical, and microbiological markers of menopausal ageing facial skin. The sketch showing the entire procedure of a cosmetic product’s preparation and testing is reported in Figure 2.

## 2. Materials and Methods

### 2.1. Patients

The clinical study enrolled a group of 104 healthy adult Caucasian female volunteers (Fitzpatrick skin type I–III, age 46–57 years; mean age 50.6 ± 2.4 years) following the exclusion and inclusion criteria for an open single-blind clinical study. All female participants were in the pre-menopause, menopause, or post-menopause period. The inclusion criteria were set as follows: (i) healthy adult female subjects, 45–60 years of age, (ii) subjects within menopause age, (iii) subjects with visible age-related symptoms of facial skin, (iv) subjects who agreed to suspend any previous application of anti-age cosmetics and intake of antioxidant nutraceuticals/drugs for at least 1 week before the trial and during the entire duration of the same, (v) subjects without any difficulty to understand and follow the clinical investigator’s instructions. Female subjects outside the defined age limits, and/or without visible signs of ageing of facial skin, and/or with history of allergic/intolerance reactions to any component of the test cosmetic products, and/or being simultaneously engaged in other clinical trials were excluded from the study. The enrolled subjects were informed that they could interrupt the clinical trial at any moment without obligation to provide a causative reason, or in case they noticed any adverse reaction to the test products, or felt in any way that the test-product application was negatively affecting their appearance.

The protocol of the clinical trial was duly analysed and approved by the local Ethical Committee (Dermatology Department; Microbiology, Immunology, and Virology Department—Kabardino-Balkar State Medical University named by Berbekov, Protocol number—N47-SI/2023). All recruited subjects gave their informed consent to personal and anamnestic data and biological material collection. The guidelines of the Helsinki Declaration for human experimentation were strictly followed throughout the whole duration of the clinical trial.

### 2.2. Test Cosmetic Products and Their Actives

Four cosmetic serums with rejuvenation-associated claims were developed and produced in compliance with the Good Manufacturing Practice regulations by Medena AG, Affoltern-am-Albis, Switzerland. As active ingredients, extracts of patented concentrated homogenates of meristem cells derived from the medicinal plants *Leontopodium alpinum* (serum n1)*, Buddeleja davidii* (serum n2)*, Centella asiatica* (serum n3)*,* and *Echinacea angustifolia* (serum n4) were used. These water–glycerin extracts were purchased from (I.R.B. Spa by Croda International Plc, Vicenza, Italy) and used in the concentrations recommended by the producer, namely, *Leontopodium alpinum* (1%), *Buddeleja davidii* (1.2–1.4%), *Centella asiatica* (1.8%), and *Echinacea angustifolia* (0.5%). The additional active ingredients claimed by the cosmetic manufacturer were hyaluronic acid/sodium hyaluronate and panthenol to invigorate the moisturising effect, as well as Alpine Glacier Water, with lower-than-normal content of the heavy isotope of hydrogen D, deuterium (142 ppm D in Alpine Glacier Water vs. 156 ppm D in the distilled tap water), capable of reducing D-content in both the cells and extracellular matrix of the ageing skin [11]. Basic compositions of the manufacturer’s property were used as excipients. The cosmetics line is marketed and distributed by Swiss Image International AG, Zug, Switzerland, under the proprietary trademark SWISS IMAGE Infinity Serums.

### 2.3. Clinical Study Design

The entire trial duration was set as 4 weeks. One hundred and four healthy females in the menopause period were recruited in accordance with the inclusion and exclusion criteria specified under Section 2.1*,* after obtaining their informed consent to participate in the randomised single-blind clinical study. The protocol included two visits by the dermatologists selected as investigators for this trial. At the 1st visit, the test cosmetic products were distributed in the quantity necessary for the entire period of 4-week applications to all participants, who were carefully instructed on how to apply the products. Two randomly selected serums out of the 4 Infinity Serums were assigned to each participant for daily application, respectively, in the morning and at bedtime. In detail, each person applied either n1 + n2 serums, n1 + n3 serums, n1 + n4 serums, n2 + n3 serums, n2 + n4 serums, or n3 + n4 serums. The questionnaire for the clinical assessment of the effects of cosmetic treatment was developed and filled up by dermatologist investigators (Doctor’s Questionnaire). The facial skin parameters and the ophthalmological status of the enrolled subjects were analysed twice: at the 1st visit (enrolment) and at the 2nd visit immediately after the treatment period (week 4). Each assessment session comprised instrumental methods for measuring skin physiology parameters and ultrasound properties of the skin layers. All participants were asked to consent to the collection of sebum (skin surface lipids) samples from five different parts of the face for further microbiological and biochemical analyses. A representative subgroup of participants (n = 12) equally belonging to all treatment subgroups gave their informed consent to donate skin samples of facial skin by mean of slight scarification from hidden frontal areas. These samples were used for hydroxyproline content determination.

### 2.4. Clinical Safety and Aesthetic Efficacy Assessment

The female subjects pre-selected according to the defined eligibility/exclusion criteria reached the clinic prior to the baseline visit date to read and sign Informed Consent Forms and fill out a brief personal history/medical history summary. Participants arrived at the baseline appointment (1st visit) with clean face skin without make-up, having complied with all trial requirements regarding the application of cosmetics and the topical/systemic use of antioxidants in the previous week. They were examined by expert dermatologists/cosmetologists for the visual assessment of the grade of skin dryness, redness and visible capillaries (rosacea), peeling/scaling, pigmentation, intensity of pigmentation, skin dullness, hydration, elasticity, static wrinkles, wrinkle depth, acne/post-acne defects, and pores’ enlargement. The ophthalmological control carried out by a clinician expert in eye diseases was assisted by a slit lamp, allowing them to observe frontal eye areas, eyelids, conjunctiva, iris, lens, sclera, and cornea. The ophthalmological evaluation included the occurrence of eye burning, pinching, itching, redness, tearing, changes to the eyelid skin, and oedema of the eyelids.

The objective observations of the doctors (dermatologist/cosmetologist and ophthalmologist) were registered by the Doctor’s Questionnaire at the 1st and the 2nd visits (28 days after treatment start). The following skin defects were assessed: changes in the facial skin relief, pallor degree, colour heterogeneity, scaling, static wrinkles, facial skin folding, photo-ageing degree, pigmentation, redness, and dryness.

The compliance to treatment, its safety, and the subjective opinion of the participants on the cosmetic effects and on the organoleptic properties (quality) of the test products were assessed by means of the Patient’s Questionnaire, which was filled up twice (at trial beginning and cessation). The following questions were answered: pruritus, tightness, dryness, rough surface, redness, irritation, colour heterogeneity, brightness, smoothness, shininess, wrinkling, and general appearance.

The answers to both questionnaires were scored, and a mean value of all answers was calculated. Questionnaires were evaluated by the clinical investigators to ascertain if any potentially adverse effects had occurred in the course of the trial.

### 2.5. Instrumental Assessment of Skin Physiology Parameters

Several physiological parameters, mainly describing the barrier properties of facial skin were assessed by appropriate SoftFX Skin Analyser probes ((http://www.callegari1930.com/documents/DEPLIANT_SOFT_FX_DIGITALE_ENG.pdf accessed on the 5 August 2024), Callegari Srl., Parma, Italy), with high-resolution micro-photo-camera (CAM FX, Parma, Italy) visual analysis and patented computerised programs. Skin elasticity was determined by the elastometric approach used in the ELA FX technique (stress/deformation of skin by suction application) and expressed in arbitrary units (u.a.), and sebum content was measured by the TAPE FX sebum probe (computer vision processing of sebum-sensitive tape). A computerised multifunctional diagnostic tool of the SoftFX CallegariSrl., Parma, Italy, employing a calculation algorithm which integrates different objective morphometric parameters (epidermal thickness, tone, wrinkles, moisture, sebum levels, pigmentation sport, and elasticity), was employed for the assessment of the face skin biological age index (Callegari Srl., Parma, Italy). The elastic properties of the skin were additionally analysed by a TPM system containing an elastometry sensor (22 MHz), which combines digital ultrasound examination with an imaging record (DUB CUTIS, Hamburg, Germany).

The trans-epidermal water loss (TEWL), an index of epidermal water barrier status and, consequently, an indirect index of skin moisture levels, is an objective quantitative measurement of skin integrity, measured as the amount of water lost (evaporated) across the stratum corneum per area of skin, assessed by means of a Tewameter^®^ TM300 (Courage+Khazaka Electronic GmbH, Germany) and expressed as g/h/m^2^. When the epidermis is aged or damaged, the barrier properties of the skin are affected, with increased water evaporation and consequently reduced skin hydration. The hydrating and smoothing effects of the cosmetic products were assessed by a corneometry method using the Corneometer^®^ CM 820 (Courage+Khazaka Electronic GmbH, Germany). The apparatus determines water content in the upper layer of the skin with depth equal to 30 μm.

### 2.6. Assessment of the Ultrasound Properties of the Skin

The assessment of the ultrasound properties of the skin was performed by means of a digital ultrasound imaging system DUB CUTIS (Digital Ultraschall Bildsystem, Germany), which allows the simultaneous determination of four parameters: epidermal and dermal thickness and epidermal and dermal ultrasonic density. The former two parameters represent indirect markers of collagen (dermis) and lipid (epidermis) synthesis and retention, while the latter pair characterises the evenness and order in the epidermal and dermal structures, respectively.

### 2.7. Assessment of Direct Anti-Microbial Effects of the Test Cosmetics and of the Skin Microbiota Pattern

Two types of microbiological tests were carried out: (1) the Agar-Well Diffusion Test was used to determine the possible direct anti-microbial effects of the mixture (1:1:1:1 *v*/*v*) of the Infinity Serums; (2) microbiological analyses of the skin before and after the application of all serums (2 randomly selected serums out of 4 serums under investigation per each participant) were performed by a quantitative real-time polymerase chain reaction (RT-PCR).

(1)A bacterial and yeast cell suspension obtained after overnight culture was spread uniformly on the solid agar medium and left dried at room temperature. The wells were cut using a sterile Pasteur pipet, and the diameter of the wells was the same in each experiment (8 mm). Then, 50 μL of the serum mixture was loaded and kept in chilled conditions for 2 h to allow diffusion into the agar. Then, another 50 μL aliquot of the tested serum mixture was added to the wells. A potassium buffer solution (PBS, pH = 7.2) was used as a negative control, and a mixture of the antibiotics penicillin, streptomycin, and acetic acid was used as a positive control, for bacteria and yeast, respectively. The agar plates were incubated at 37 °C for 24 h for *Staphylococcus aureus*, *Propionbacterium acnes*, *Escherichia coli*, *Staphylococcus epidermidis*, and *Candida albicans*. A clear zone diameter around the well, indicating microbial inhibition, was measured at two perpendicular directions. The minimal inhibitory concentration (MIC) was determined by testing different volumes of the mixture diluted by PBS. All measurements were performed in triplicate [48].(2)Microbiological analyses of the facial skin swabs collected from 12 randomly selected participants were performed by a quantitative real-time polymerase chain reaction (RT-PCR) (differential bacterial counting). For RT-PCR tests, DNA was isolated from samples of skin swabs taken by sterile water-soaked cotton sticks and kept on ice for no more than 12 h. DNA was amplified with iQTM Supermix using the MiniOpticon Real-Time PCR Detection System (Bio-Rad, Hercules, CA, USA). All real-time assays were carried out under the following conditions: 35 cycles of denaturation at 95 °C for 15 s; annealing and extension at 60 °C for 60 s. Melt curve analysis was performed to confirm the specificity of the amplified products. All samples were run in triplicate, and relative expression was determined by normalising samples to housekeeping genes [49].

### 2.8. Hydroxyproline Assay

The skin levels of free hydroxyproline (Hyp) and Hyp in the form of oligopeptides, mainly proline-hydroxyproline, were determined on the hidden frontal skin scarification samples by a chemical colorimetric method using a commercial kit (Hydroxyproline Detection Kit, Bio-Rad, Hercules, CA, USA) following manufacturer’s instructions. Hyp concentrations were quantified in the linear range of its calibration curve using an array reader, and results were expressed in μg/mg of protein.

### 2.9. Assessment of Lipid Peroxidation in Skin Lipids

The biochemical determination of oxidation products in samples of skin sebum quantitatively taken by sterile, ethanol-soaked cotton sticks was carried out by the classical determination of malonyldialdehyde (MDA) content. In brief, 100 μL of egg yolk was mixed with 10 μL of facial skin sebum extract. Then, 100 μL of FeSO_4_ was added, and the volume was adjusted to 1 mL with physiological solution. The mixture was incubated at room temperature for 30 min, and 0.5 mL of 20% trichloroacetic acid plus 0.1 mL of 0.01 M butyl hydroxy toluol in ethyl alcohol were added. The tubes were centrifuged at 1500× *g* for 10 min, and supernatant was collected. The mixture of 0.7 mL of supernatant and 0.6 mL of 0.5% thiobarbituric acid (TBA) was heated at 100 °C for 30 min and cooled down, and the absorbance at a wavelength of 532 nm was determined. The intensity of lipid peroxidation was expressed as MDA content (μM/L of skin sebum lipids [50].

### 2.10. Statistical Evaluation

Clinical data were analysed using WINSTAT programs for personal computers (Statistics for Windows 2018, Microsoft, CA, USA). Unpaired *t*-test and Mann–Whitney test were carried out to make comparisons between the beginning and cessation of the trial, reporting mean percent improvement from baseline, *p*-value vs. baseline, and PT-value vs. intermediary 10-day term. Within-group differences were evaluated with a paired *t*-test (mean difference from baseline) and standard descriptive statistics (mean, median, standard deviation S.D. and standard error S.E.M.). Statistical significance was indicated at *p* ≤ 0.05.

## 3. Results

### 3.1. Dermatological and Ophthalmological Safety of the Test Cosmetics

One hundred and two recruited females completed the study. Two drop-outs occurred due to personal reasons not connected to low compliance or adverse effects of the test cosmetics. The products were well tolerated by the study panel (100%) with no adverse reactions reported during the study trial period, as could be concluded from both Doctors’ and Patients’ Questionnaire Scores (0—no problem; 1—weak problem; 2—moderate problem; 3—substantial problem; 4—highly expressed problem). During regular products’ application over a period of 28 days, there were no reports of negative symptoms and feelings, such as skin redness, swelling, burning, tingling, irritation, or itching, that might indicate an intolerance to any product or to any component of the products. In accordance with the Patients’ Questionnaire, the average summary score of all complaints was diminished from 1.2 ± 0.3 at the beginning of the trial to 0.5 ± 0.2 at its cessation (*p* < 0.05). According to the Doctors’ Questionnaire, the baseline score 6.7 ± 1.1 dropped to 3.4 ± 0.6 (*p* < 0.01) at week 4.

The ophthalmological evaluation showed no changes in the anterior segment of the eye and in the eye-protective apparatus in all 102 participants successfully completing the study. In these females, the test products did not cause eye burning, pinching, itching, redness, tearing, changes in the skin of eyelids, or oedema of the eyelids.

### 3.2. Effects of the Test Cosmetics on the Facial Skin Physiology and Ultrasonic Properties

The analyses of the main physiological parameters of the skin relevant to ageing, such as smoothness, moisture (Figure 3), elasticity (Figure 4), TEWL, and sebum content, were statistically significantly improved after test cosmetics’ application for 4 consecutive weeks. All these instrumental data are presented in Table 2. From them, the index of biological age of facial skin was calculated using the manufacturer’s formula. Its value statistically significantly changed in the direction of a younger biological age.

The ultrasonic examination of the facial skin in five different points, such as eye corners, cheeks, and forehead, by the ultrasonic device before and after the study treatment showed a significant difference in the images taken for the same patient (Figure 5). The measurements of the density within the dermis and epidermis layers revealed statistically significant changes in the dermal ultrasonic parameters, i.e., the thickness of the dermal layer and the acoustic density of derma, reflecting an increase in the collagen fibres content and an improvement in collagen network structure (see green areas, Figure 5), respectively (Table 3), while there were no significant changes in the epidermis (Table 4).

### 3.3. Effects of the Test Cosmetics on the Biochemical Markers of Skin Ageing

The content of hydroxyproline—an amino acid known as a unique marker for collagen—was determined at the beginning and end of the study to confirm the positive changes observed in the facial skin dermal layer, where collagen is the major structural component defining the skin elasticity, smoothness, and plumpness characteristic of younger skin. A limited number of participants (n = 12) volunteered themselves to donate a very small amount of scarified skin from forehead or cheeks. The skin cells and extracellular matrix were processed as described in Materials and Methods, and measurements were carried out (Figure 6). A clear-cut (*p* < 0.0068) increase in the content of hydroxyproline, an indicator of the enhanced collagen synthesis, was observed.

Another extremely important marker of ageing (chronologically or/and premature) is the intensity of lipid peroxidation in the skin sebum lipids. Lipid peroxidation is driven by oxygen or/and nitrogen reactive species and is suppressed by endogenous cutaneous antioxidants or antioxidant substances supplemented by facial cosmetics/cosmeceuticals [51,52,53]. It is common knowledge that more mature skin is less protected by natural endogenous antioxidants, which are decaying with age, due to hostile environmental conditions, bad habits, hormonal status, and skin pathologies [51,54,55,56]. Here, we measured levels of MDA, an end product of lipid peroxidation, in the facial skin lipids (Table 5). The regular application of the Infinity Serums resulted in a remarkable drop of MDA content (*p* < 0.01).

### 3.4. Direct Anti-Microbial Effects, and Changes in the Facial Skin Microbiota after the Regular Four-Week-Long Application of the Test Cosmetics

In the first step, the evaluation of the Infinity Serums’ effects on skin microbiota was carried out by traditional microbial counting of the number of pathogenic microbes characteristic of facial skin and by the determination of the minimal inhibitory concentration (MIC) of the mixture of all four serums taken in the volume ratio 1:1:1:1. The results are shown in Table 6. The mixture of serums exerted rather strong antibiotic-like effects towards skin pathogens.

The applications of the Infinity Serums regularly for 4 weeks did not greatly affect the symbiotic pattern of the facial skin microbiota. According to real-time PCR tests performed for a randomly selected group of participants, the content of 48 microbes characteristic of human facial skin was not changed significantly (Appendix A, Table A1). Interestingly, four microbial species associated with ageing skin problems/certain skin pathologies were found suppressed at the trial cessation (Table 7).

## 4. Discussion

Moving away from using animal-derived products, experts in aesthetic medicine have begun to discuss the feasibility and clinical efficacy of higher plant- and marine-organism-derived active agents. Plants, unlike humans and animals, do not have immune, nervous, and endocrine systems to protect themselves against the unfriendly environment, infections, wounding, herbivores, insects, tumours, and rival species. In the course of evolution, plants have evolved one extremely efficient mechanism to protect themselves from these biotic and abiotic stresses—they synthesise secondary metabolites [8,9,10,11]. Secondary metabolites are small molecules like non-protein hormones neurotransmitters, or mediators of immune response in human beings. Their parent molecule is the amino acid phenylalanine. The same amino acid is a parent molecule for human neurotransmitters and hormones, such as adrenalin, noradrenalin, serotonin, thyroid hormones, oestrogens, androgens, etc. Due to this unique parent, definite secondary plant metabolites are highly available to human cells, binding to the same parts of human cellular membranes (receptors) as the molecules of human origin. They are also perfectly metabolised in human cells through similar enzymes and metabolic pathways (high compatibility). Secondary plant metabolites can imitate numerous biological actions of human regulatory molecules, thus exerting enhanced biological effects on human cells. Plant-derived secondary metabolites derived from the shikimic acid/amino acid phenylalanine with a potential for cosmetic industries are shown in Figure 7.

The industrial production of secondary plant metabolites is limited by a number of factors: (1) low abundance in natural sources/extinguishing natural species; (2) seasonal variations in plant harvesting; (3) contamination of plants by environmental pollutants (e.g., heavy metals, oil- and gas-derived toxic hydrocarbons, pesticides, aflatoxins, mycotoxins, and other organic and non-organic hazardous compounds); (4) complex and expensive procedures for the extraction and purification of the actives from the grown plant tissues; (5) poor standardisation of the final product due to unavoidable variations in soil, water, weather conditions, and fertilisers used for plant growth.

In an attempt to find out valid alternatives to the production of desirable natural compounds, plant cell culture technologies/meristem plant cells are being developed as a valuable source of high-quality plant-based medicinal/cosmetic ingredients [16,17,20,21,57] for sustainable mass production. For example, cultured meristem cells exposed to appropriate stimulators (elicitors) produce easily isolated and uncontaminated modulators of oxidative state (free radical scavengers, direct and indirect antioxidants) [58], UV + visible + infrared light protectors [30], chelators binding toxic heavy metals [59], and anti-bacterial or anti-inflammatory agents [19,20,21,22,23,60,61].

In the present clinical laboratory study, significant improvement of menopause-associated physiological peculiarities of the facial skin, such as skin dryness/loss of moisture, loss of smoothness (Figure 3) and elasticity (Figure 4), impaired barrier function, TEWL, and inhibited sebum production, was observed so that the index of face skin biological age calculated from these parameters was found to be improved vs. before the trial (Table 2). There are data in the literature showing that the actives of meristem plant cells (verbascoside, echinacoside, asiaticoside, and leontopodic acids) used in the Infinity Serums possess similar properties (See references of Table 1). According to several publications, these secondary metabolites are able to induce collagen production by human skin fibroblasts in vitro [26,31,45,47]. Here, we showed, for the first time, that serums containing the above mentioned secondary metabolites increased the content and improved the network structure of facial skin collagen in real-time human application (Figure 5, Table 3 and Table 4). Moreover, biochemical tests on the content of hydroxyproline as a unique marker of collagen confirmed quantitative ultrasonic data on the highly enhanced collagen content in the dermal but not in the epidermal layer of the skin (Figure 6). It is of common knowledge that skin fibroblasts—cells producing collagen—are located in the dermal compartment of the skin. The molecular mechanism(s) of collagen-stimulating action by active biomolecules from concentrated plant meristem cells still has to be evaluated.

Regarding the inhibition of lipid peroxidation in the skin lipids as revealed by the content of MDA, a final product of free-radical-driven lipid peroxidation (Table 5), this could be caused by a combination of different mechanisms supported by the bioactive molecules in the facial cosmetics under investigation. Practically all of them are direct scavengers of free radicals/active oxygen species initiating lipid peroxidation in the extracellular lipid mantel of the skin as well as in the skin cell membranes (See references in Table 1, [21]). Verbascoside and its close metabolite teupolioside, major secondary metabolites in the *Buddleja davidii* and *Syringa vulgaris* meristem plant cells, are also known as a potent chelator of Fe^+2^, a catalyst of the reaction of lipid peroxidation [59]. Verbascoside and leontopodic acids induce the expression of endogenous antioxidant enzymes through the nuclear factor erythroid 2–related factor 2 (Nrf2)-dependent mechanism, thus stimulating the natural antioxidant defence of the skin [22,58] (Figure 8). Moreover, verbascoside has been shown to rescue cutaneous non-enzymatic lipid-soluble sacrificing antioxidants, such as vitamin E, squalene, coenzyme Q10, and sterols [23]. The sketch of possible endogenous cutaneous targets to diminish lipid peroxidation in the skin induced, for example, by solar radiation is presented in Figure 8.

The extremely important role of skin microbiota—a certain pattern of symbiotic and pathogenic microorganisms residing on human skin—in skin and general organism immunity, skin susceptibility to environmental and intentional man-made factors, and skin chronological and premature ageing have been discovered and intensely evaluated during the last 1–2 decades [10,62,63]. On these grounds, many emerging cosmetics have started claiming microbiota-modifying effects referring mainly to the information of manufacturers of raw materials. The clinical data on the effects of cosmetic compositions/products are scarce, and they appear practically exclusively in the company brochures and not in reliable peer-reviewed publications. As a first step, we checked the direct anti-microbial action of the Infinity Serums in the bacterial and yeast cultures of microbes relevant for facial human skin physiology and pathology (Table 5). It was found that all the cosmetic serums studied exhibited slight antibiotic-like effects (lysis area and minimal inhibitory concentration). Similar data have been published previously for verbascoside/teupolioside against *Malassezia furfur* [59]. Application of the Infinity Serums for one month did not affect general microbiota patterns as determined by the real-time PCR test, as 48 out of 55 microbial species remained unchanged (Table A1, Appendix A). However, there were four microbes whose role in the pathogenesis of menopause-associated *acne vulgaris* or *candidosis* [1,2,3,5] was significantly suppressed by the application (Table 6). We assume that the microbiological data confirm that the ready cosmetics preserved the antibiotic-like effects of the raw materials—pure actives from meristem plant cells. On the other hand, they allowed for maintaining the symbiotic balance without making poorer the microbiota pattern, which is clearly considered as a negative outcome.

## 5. Conclusions

This open case–control randomised clinical study of cosmetics containing extracts of cultured meristem plant cells as actives was carried out to demonstrate safety and aesthetic effects on the facial skin of women in pre-menopausal, menopausal, and post-menopausal periods. Evident limitations of this clinical trial such as the absence of placebo control could be easily attenuated, because dermatological/cosmetic outcomes were measured not only subjectively (by patients) or by clinical observations (doctors), but mainly instrumentally, and were confirmed by biochemical and microbiological tests. The major active ingredients of meristem plant cells are secondary metabolites, such as leontopodic acids, verbascoside, asiaticoside, and echinacoside, with well-known biological activities towards skin cells and the extracellular matrix. However, clinical data on their effects towards markers of menopausal skin are absent. To the best of our knowledge, this is the very first clinical study showing statistically significant changes in the skin physiology markers characteristic of ageing/menopause (elasticity, barrier functions, moisture, sebum, ultrasonic properties, and collagen content and structure). The instrumental, biochemical, and microbiological data confirm that the cosmetics shifted the skin physiology parameters to a younger biological age, enhanced collagen synthesis, inhibited lipid peroxidation, and favoured normal microbiota. It seems that cultured meristem plant cells could be a unique ethical and ecologically friendly source of actives for clinically and aesthetically effective anti-age cosmetics.

## Figures and Tables

**Figure 1 biomolecules-14-01176-f001:**
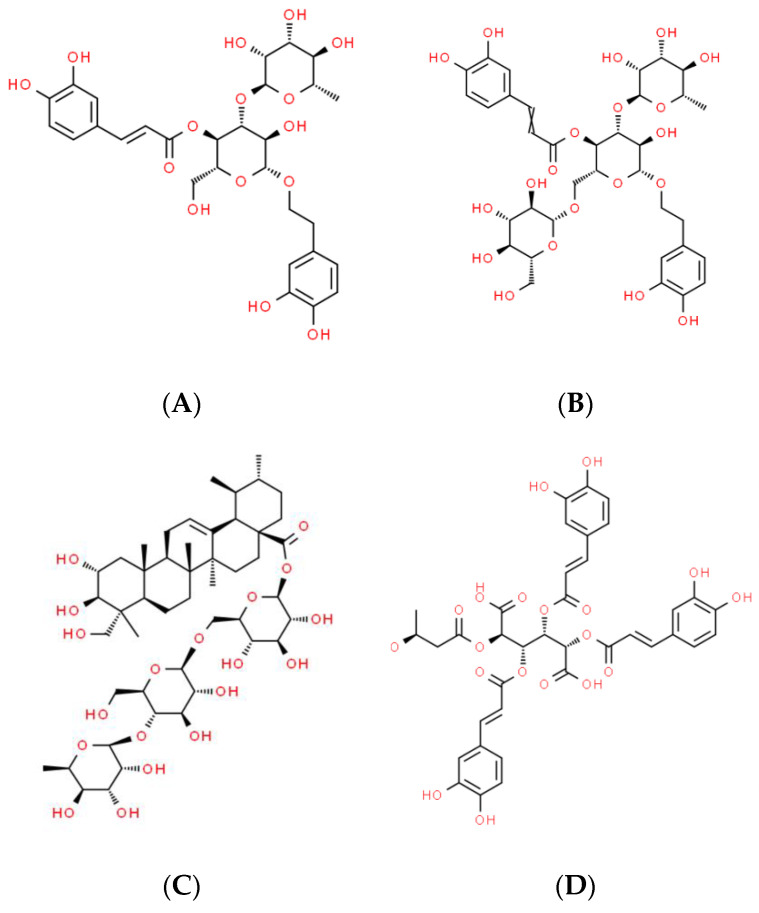
Chemical structures (retrieved from www.ChemSpider.com; accessed on 18 April 2024) of the major actives identified in the plant meristem cell concentrates used in the cosmetics in question. (**A**)—verbascoside (glycosylated phenylpropanoid); (**B**)—echinacoside (glycosylated phenylpropanoid); (**C**)—asiaticoside (triterpenoid glycoside); (**D**)—leontopodic acid (glycosylated phenylpropanoid-like).

**Figure 2 biomolecules-14-01176-f002:**
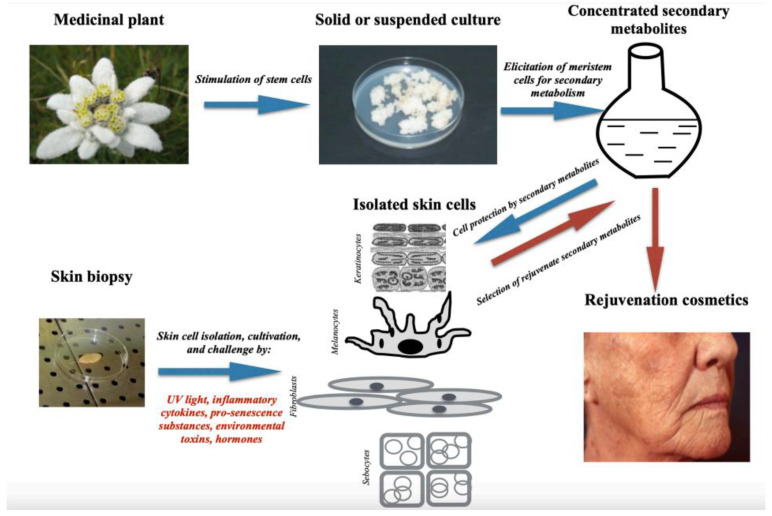
Production and pre-clinical testing of rejuvenation cosmetics with meristem plant cell-concentrated extracts as active ingredients.

**Figure 3 biomolecules-14-01176-f003:**
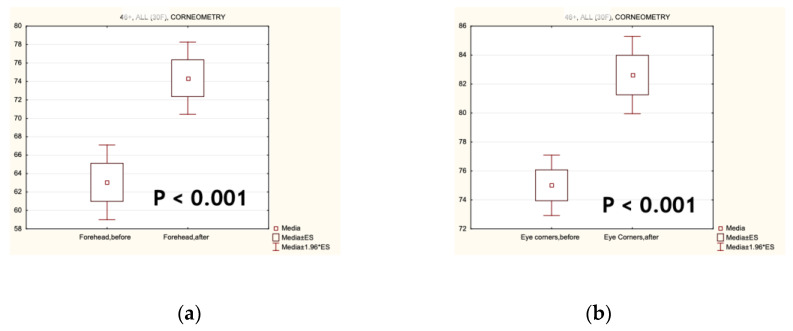
Hydrating and smoothing effects of the Infinity Serums on the facial skin of forehead (**a**) and eye corners (**b**) (n = 102).

**Figure 4 biomolecules-14-01176-f004:**
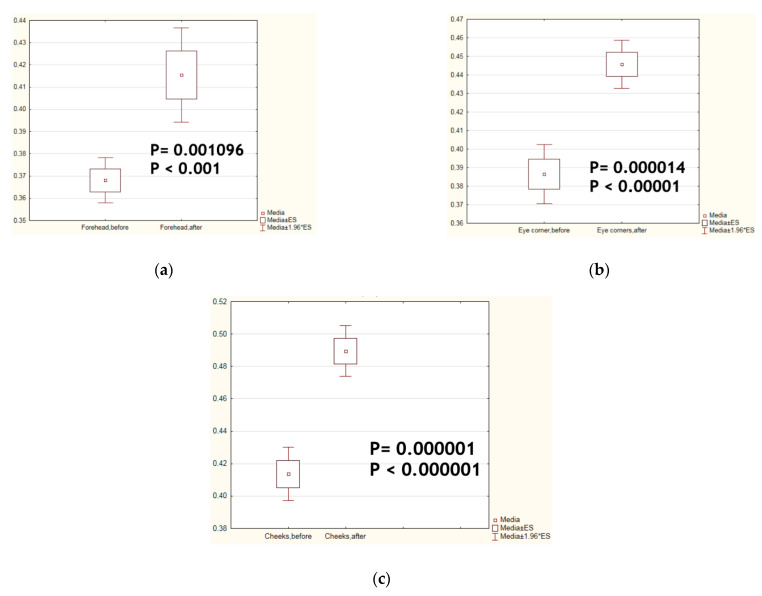
Changes in the facial skin elasticity after Infinity Serums’ application for 4 weeks. (**a**) forehead; (**b**) eye corners; (**c**) cheeks (n = 102).

**Figure 5 biomolecules-14-01176-f005:**
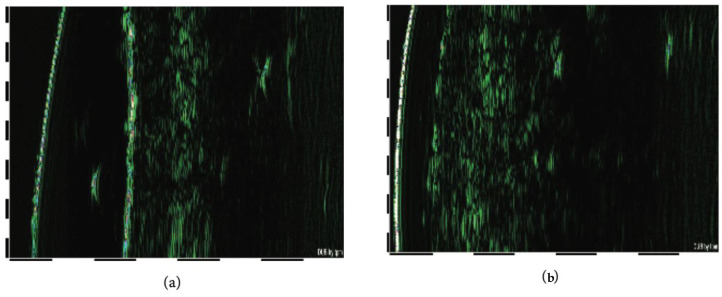
Ultrasound images of the skin before (**a**) and after (**b**) the serum application for 4 weeks.

**Figure 6 biomolecules-14-01176-f006:**
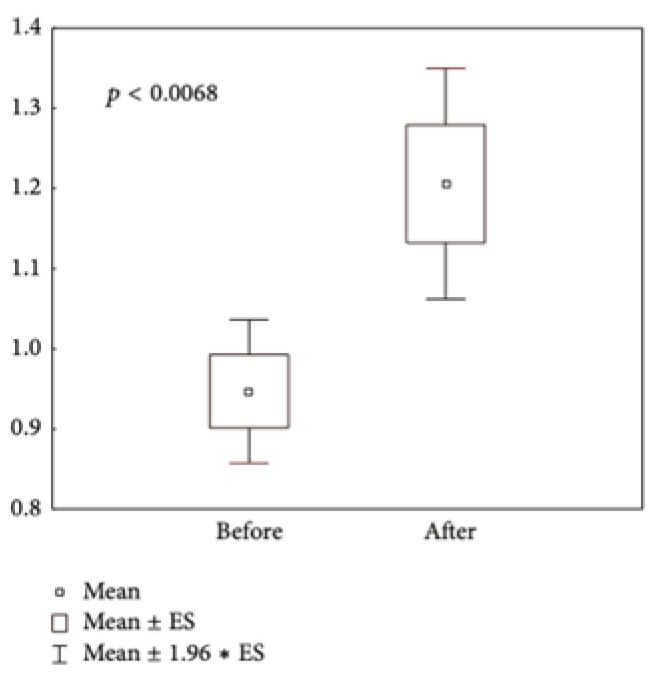
Levels of skin hydroxyproline (μg/mg protein) before and after a one-month treatment with the Infinity Serums in a selected group of patients (n = 12).

**Figure 7 biomolecules-14-01176-f007:**
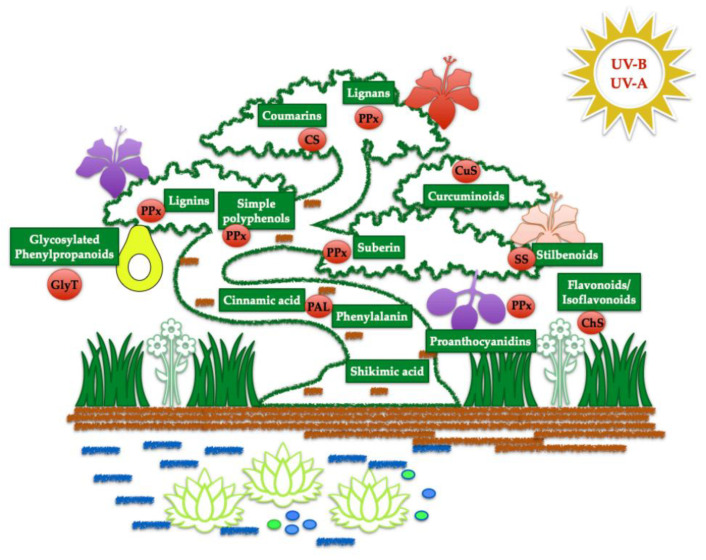
Metabolic pathways to synthesise secondary higher plant metabolites from shikimic acid-phenylalanine. Higher plants, being exposed to biotic (infections, rival species, insects, herbivors) and abiotic (solar radiation, high/low temperatures, changes in soil and air components) stresses, start defensive metabolic responses as a chain of enzymatic and non-enzymatic reactions resulting in secondary polyphenols. Abbreviations for enzymes: PAL—phenylalanine liase; PPx—polyphenol peroxidases; GlyT—glycosyl transferase; ChS—chalcone synthase; CuS—curcumine synthase; CS—coumarin synthase; SS—stilbene synthase. Marine organisms/plants (lower part of the figure) evolved different metabolic pathways in response to stresses.

**Figure 8 biomolecules-14-01176-f008:**
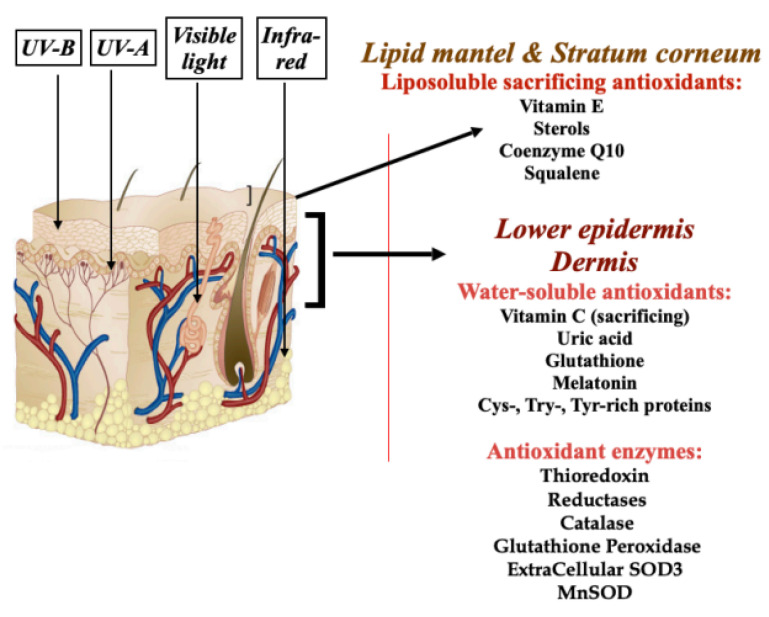
Endogenous cutaneous antioxidant barrier against solar light as a target for rejuvenation cosmetics. This very complex defensive system consists of non-enzymatic lipid-soluble and water-soluble antioxidants, the content of which declines with chronological age and in premature ageing. These antioxidants could be replenished by cosmetics packed with antioxidants delivered to definite skin targets by an optimised vehicle. Cosmetics containing biologically active substances are able to rescue lipid-soluble sacrificing antioxidants from oxidation by environmental factors, bad habits, and certain pathologies. Biomolecules of advanced cosmetics are potent inducers of endogenous cutaneous antioxidant enzymes through an Nrf2-dependent mechanism.

**Table 1 biomolecules-14-01176-t001:** Plant meristem cell actives targeting human skin cells in vitro and in vivo.

Plant Meristem Cells	Active Compound IUPAC Name	Effects on Human Skin
*Buddeleja davidii*	Verbascoside 2-(3,4-Dihydroxyphenyl)ethyl α-L-rhamnopyranosyl-(1→4)-{5-*O*-[(2*E*)-3-(3,4-dihydroxymethyl)prop-2-enoyl]-β-D-glucopyranoside}	UVA, UVB protection [15,16,17] Anti-senescence factors for skin cells in culture [18] Anti-ageing [19] Anti-inflammatory [17] Direct and indirect antioxidants via NRF2 pathway [17,19,20,21,22] Rescue of endogenous antioxidants [23]
*Leontopodium alpinum*	Leontopodic acids A & B 3,4,5-Tris-O-[(2E)-3-(3,4-dihydroxyphenyl)-2-propenoyl]-2-O-[(3S)-3-hydroxybutanoyl]-D-glucaric acid	Antioxidant [24,25] Blue-light-induced damage preventing [24,26] Collagen destruction preventing [26] Anti-inflammatory [27] Anti-chronological ageing [28]
*Echinacea angustifolia*	Echinacoside 2-(3,4-Dihydroxyphenyl)ethyl α-L-rhamnopyranosyl-(1→3)-[β-D-glucopyranosyl-(1→6)]-β-D-glucopyranoside 4-[(2*E*)-3-(3,4-dihydroxyphenyl)prop-2-enoate]	Anti-photo-ageing by inhibition of oxidative stress and DNA damage [29,30] Anti-photo-damage through inhibition of collagen degradation [31] Anti-inflammatory through TNF pathway suppression [32] Anti-inflammatory and wound healing promoting [33] Anti-angiogenesis via VEGF suppression [32] Cytoprotection from oxidative damage and xenobiotics by NRF2 induction [33] Anti-glycation [34]
*Centella asiatica*	Asiaticoside 6-Deoxy-α-L-mannopyranosyl-(1->4)-β-D-glucopyranosyl-(1->6)-1-O-[(2α,3β)-2,3,23-trihydroxy-28-oxours-12-en-28-yl]-β-D-glucopyranose	Acne, burns, atopic dermatitis, wounds [35,36] Hypertrophic scars [36,37] Photo-ageing skin, cellulite [37,38] Anti-inflammatory, atopic dermatitis [39] Wound healing via accelerated skin cell migration [40,41] Diabetic wound healing [7,42] Anti-chronological ageing/anti-wrinkles [41,43,44,45] Periorbital hyperpigmentation [46] Skin hydration, collagen synthesis [45,47]

**Table 2 biomolecules-14-01176-t002:** Physiological parameters of the facial skin (n = 102) before and after Infinity Serums’ application for 4 weeks. * *p* < 0.05; ** *p* < 0.01; *** *p* < 0.001.

Parameter, Unit	Before Treatment	After Treatment
Elasticity, Arb. Units	33.66 ± 1.21	40.26 ± 0.87 **
Smoothness, Arb. Units	12.15 ± 0.24	18.01 ± 0.37 ***
Moisture, Arb. Units	49.03 ± 3.52	56.86 ± 3.02 **
TEWL, g/h/m^2^	6.3 ± 0.03	4.5 ± 0.07 **
Sebum, Arb. Units	29.27 ± 4.76	56.85 ± 4.04 ***
Skin biological age, Years	51.5 ± 0.4	46.3 ± 1.0 *

**Table 3 biomolecules-14-01176-t003:** Effects of a 4-week treatment with the Infinity Serums on the ultrasonic properties of the dermis (n = 102). * *p* < 0.05.

Parameter	Before Treatment	After Treatment
Thickness, μm	3900 ± 31	4133 ± 28 *
Acoustic density, Arb. Units	5.1 ± 0.2	6.3 ± 0.1 *

**Table 4 biomolecules-14-01176-t004:** Effects of a 4-week treatment with the Infinity Serums on the ultrasonic properties of the epidermis (n = 102).

Parameter	Before Treatment	After Treatment
Thickness, μm	77.0 ± 0.8	77.6 ± 0.9
Acoustic density, Arb. Units	35.2 ± 2.2	35.4 ± 2.0

**Table 5 biomolecules-14-01176-t005:** Content of the lipid peroxidation product MDA in the facial skin lipids (n = 24). ** *p* < 0.01.

Cosmetics	Number of Participants (Number of Sebum Samples)	Duration of Application (Days)	MDA (μmol/L Sebum)	
			Before	After
All Infinity Serums	24 (144)	28	26 ± 4	15 ± 1 **

**Table 6 biomolecules-14-01176-t006:** Direct anti-bacterial effects of the mixture of Infinity Serums (1:1:1:1, *v*/*v*) in the skin pathogen cultures.

Pathogen	Negative Control	Antibiotic Control	Mixture of Infinity Serums	
	Inhibition Zone (mm)	Inhibition Zone (mm)	Inhibition Zone (mm)	MIC Values (μL)
*Propionbacterium acnes*	0	10.22 ± 1.31	6.15 ± 0.97	50
*Staphylococcus aureus*	0	7.68 ± 0.87	4.87 ± 1.11	60
*Streptococcus epidermidis*	0	9.66 ± 0.79	5.60 ± 1.23	60
*Esherichia coli*	0	8.32 ± 0.43	3.88 ± 1.54	75
*Candida albicans*	0	10.89 ± 1.59	6.65 ± 2.04	25

**Table 7 biomolecules-14-01176-t007:** Microbes/pathogens on the facial skin before and after Infinity Serum application for 4 weeks.

Microbes	Normal Values, 10^5^ cells/g	Results, 10^5^ cells/g		
		Before the Trial	After the Trial	*p*, Before/After
*Streptococcus* spp.	249 ± 45	205 ± 50	160 ± 25	<0.05
*Propionbacterium acnes*	0–42	50 ± 5	10 ± 3	<0.05
*Propionbacterium freudenechii*	3480 ± 470	4200 ± 470	2980 ± 200	<0.05
*Propionbacterium jensenii*	38 ± 3	38 ± 6	28 ± 3	<0.05

## Data Availability

Data are available as hard copies in the archives of the Microbiology, Immunology, and Virology Department of Kabardino-Balkar State Medical University named by Berbekov. They could be provided upon request.

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
