# Peer review of "Effects of Plant Meristem-Cell-Based Cosmetics on Menopausal Skin: Clinical Data and Mechanisms"

_biomolecules, 2024, doi:10.3390/biom14091176_

Round 1

Reviewer 1 Report

Comments and Suggestions for Authors

The introduction is very general and not focused on introducing the study properly. The device used to measure the skin physiological parameters seems to be not professional (or at least not referenced in the literature). The discussion/conclusion section is too long and not give insights useful for interpreting the results. The evidence on skin microbiome effect is low. The flow of information in the manuscript is confusing the reader. A placebo group is missing (the main limitation of the study) to make the results and their discussion strong. Please find here below some recommendations to improve the manuscript.

Abstract

The abstract should summarize the main results obtained in the study instead of describing the endpoints.

Introduction

The introduction section is general and not focused on introducing properly the study.

- Line 46-78. The authors report the effects of menopause on collagen, oxidative stress, and other typical alterations related to estrogen deficiency that are not closely related to the endpoints measured in the study. I would recommend the authors focus the introduction section on the endpoints of the study.

- Line 94-98. The sentence is self-referencing and not supported by evidence. In my opinion, it should be deleted since does not add any value and is irrelevant to the overall economy of the manuscript.

- Table 1 should be moved to the supplementary material section or deleted.

- Figure 1 should be moved to the supplementary material section or deleted.

- Figure 2 should be moved to the supplementary material section or deleted or be part of the graphical abstract.

Material and methods

- Line 163-175. The ingredient list is missing. The active ingredient(s) concentration should be reported.

- Line 183-194. The sentence is not clear. Does each subject test 2 serum products or 4 serum products?

- Line 209. The list of “skin defects” is missing.

- Line 219. To the best of my knowledge the “SOFT PLUS TOP” is not a professional system used in the assessment of the efficacy of cosmetic products. Please can you add more scientific references to these probes?

- Line 221. What is the elastometric approach used in the SOFT PLUS technique?

- Line 222. TEWL is not an index of skin moisture. It is more properly related to the skin barrier effect to water evaporation (loss).

- Line 223-225. In the aged skin, the TEWL can be also decreased due to the lower skin moisture content. Less water to be evaporated > less moisture content of the skin > less evaporation ---> reduced TEWL. This is also supported by the very low TEWL values measured in the study (6.3 g/h/m2)

- Line 234. The depth of penetration of corneometer is lower than 30 mm. The depth of penetration is 10-20 mm https://www.courage-khazaka.com/en/scientific-products/corneometer-cm-825.

- Line 239-240. Epidermal thickness is not an indirect marker of lipids. It is technically impossible for ultrasounds to detect lipids. Please argue more about the relation between the parameter and the lipids

- Line 318. The meaning of the sentence “ … Its value statistically significantly changed in the direction of a younger age.” is aleatory

- Figure 3 quality is too low. I cannot understand the meaning of the figure.

- Figure 4 reports two p values. What is the difference between them?

- Table 2. The units for TEWL and sebum are missing. Please describe how the skin biological age was calculated. The skin sebum content seems to be very low.

- Figure 5. Images (a) and (b) are not comparable. The quality of the pictures is very low. Arrows should be added to explain better the image

- Table 3. The thickness of the dermis seems to be quite high.

- Line 353-359 should be moved to the introduction

- Line 365. It is not correct to speak about microbiota based on MIC results. It is too much speculative.

- Results should be discussed in detail.

- Demographic and baseline characteristics should be added

Discussion

Line 380-422. Should be partially moved to the introduction section.

Line 427. “… these parameters biological age of facial skin was found younger than before the trial” It is more correct to say "the parameters are improved". The correlation between the improvement and the younger effect is speculative and not demonstrated by data.

Figure 7 should be deleted

The data do not support the conclusion on microbiota. The microbiota-modifying effects should be further explored with more appropriate techniques.

Study limitations are missing and should be reported.

Comments on the Quality of English Language

The quality of the English language is low. Extensive editing is required. 

Reviewer 2 Report

Comments and Suggestions for Authors

Dear Authors,

The manuscript investigated the effect of facial cosmetics containing active ingredients of meristem plant cells, derived from medicinal plants Leontopodium alpinum, Buddeleja davidii, Centella asiatica, and Echinacea angustifolia on menopausal skin. In my opinion, the analyses presented throughout the manuscript are well detailed and presented. I suggest that the authors should revise the following issues before resubmitting it in order to improve both aspects of form and content. Here are some aspects that need attention:

1.     Introduction: the introduction presents too much information regarding the physiological changes that occurs in a menopausal women. The authors should highlight more why they choose these four plants and elucidate the novelty of their work in accordance with other published studies.

2.     Figure 2 should be more specific regrading the production of the cosmetic products and application on skin woman.

3.     If I understood well, the cosmetics are already marketed and distributed before the authors realized this single blind clinical study. The authors should explain why they realized this extensive clinical study after the cosmetics have already been put on the market.

4.     In the Section 2.4, the Doctor’s Questionnaire and the Patient’s Questionnaire are mentioned. The authors should describe what were the questions that the investigators asked the patients enrolled in the study.

5.     Results: I recommend the authors to present the statistical part in section 3.2 in more detail, considering the number of patients.

6.     Figure 6 is a little bit too loaded with information; the authors should redraw it.

7.     The conclusions section is missing. It should be interesting to present some future perspectives regarding these cosmetics and their use on a large scale.

Comments on the Quality of English Language

Minor eduting of English language required.

Reviewer 3 Report

Comments and Suggestions for Authors

The study “Effects of Meristem Plant Cells-Based Cosmetics on Menopausal Skin: Clinical Data and Mechanisms” is an interesting study. It is not quite clear how the authors chose to mix the four active chemicals presented in natural sources. Why did you mix two, not four active ingredients? In table 3.4 you presented the data form the serum mix of four chemicals. Please explain! How many patients had each mix? Was the mix consistent throughout the study? Total of 12 participants donated skin. What serum did they use?

Please improve the quality of figures. Please keep the font at the same size through the text.

Comments on the Quality of English Language

no comments

Round 2

Reviewer 1 Report

Comments and Suggestions for Authors

Even if the manuscript was revised, the study design is not robust due to the lack of the placebo control. The techniques applied have few references in the scientific bibliography. The study design should be reconsidered to make the study conclusion more robust. 

Comments on the Quality of English Language

Minor editing is required.

Author Response

Answer:  We added the phrase on study limitations in the Conclusions  sub chapter as per reviewer’s suggestion.

“Evident limitations of this clinical trial such as the absence of placebo control could be easily attenuated, because dermatological/cosmetic outcomes were measured not only subjectively (by patients) or by clinical observations (doctors), but mainly instrumentally, and were confirmed by biochemical and microbiological tests.”

We honestly admitted in the Abstract that: “A randomised open clinical/laboratory study was performed to evaluate safety and cosmetic efficacy of facial cosmetics for females during menopausal period.” (cyt.)  We carried out this very first clinical trial with cosmetics containing 4 different extracts of meristem plant cells  in order to prove their safety, clinical/aesthetic efficacy, as well as several mechanistic studies with human material. These products and studies were rather costly. You may notice that several laboratory measurements, such as hydroxyproline, MDA, and  RT-PCR microbiological test were done on limited number of patients. Therefore we sacrificed our strong desire to have a placebo-control  clinical data due to economical problems. We are planning next studies with one or two cosmetics based on meristem plant cells. For sure, they will be placebo- and competitor-controlled. 

The techniques applied were based on the manufacturers’ brochures/recommendations as it is acceptable in scientific publications.

As we mentioned above, our next clinical/laboratory study design will be more appropriate to  make conclusions more robust.